# Mechanochemical Synthesis of Nickel Mono- and Diselenide: Characterization and Electrical and Optical Properties

**DOI:** 10.3390/nano12172952

**Published:** 2022-08-26

**Authors:** Marcela Achimovičová, Michal Hegedüs, Vladimír Girman, Maksym Lisnichuk, Erika Dutková, Juraj Kurimský, Jaroslav Briančin

**Affiliations:** 1Institute of Geotechnics, Slovak Academy of Sciences, 04001 Košice, Slovakia; 2Synthon s.r.o., 67801 Blansko, Czech Republic; 3Faculty of Science, Pavol Jozef Šafárik University, 04154 Košice, Slovakia; 4Institute of Materials Research, Slovak Academy of Sciences, 04001 Košice, Slovakia; 5Faculty of Electrical Engineering and Informatics, Technical University, 04200 Košice, Slovakia

**Keywords:** nickel selenide, mechanochemical synthesis, planetary ball mill, nanostructured semiconductor, electrical and optical properties

## Abstract

Nickel mono- (NiSe) and diselenide (NiSe_2_) were produced from stoichiometric mixtures of powdered Ni and Se precursors by the one-step, undemanding mechanochemical reactions. The process was carried out by high-energy milling for 30 and 120 min in a planetary ball mill. The kinetics of the reactions were documented, and the products were studied in terms of their crystal structure, morphology, electrical, and optical properties. X-ray powder diffraction confirmed that NiSe has hexagonal and NiSe_2_ cubic crystal structure with an average crystallite size of 10.5 nm for NiSe and 13.3 nm for NiSe_2_. Their physical properties were characterized by the specific surface area measurements and particle size distribution analysis. Transmission electron microscopy showed that the prepared materials contain nanoparticles of irregular shape, which are agglomerated into clusters of about 1–2 μm in diameter. The first original values of electrical conductivity, resistivity, and sheet resistance of nickel selenides synthesized by milling were measured. The obtained bandgap energy values determined using UV–Vis spectroscopy confirmed their potential use in photovoltaics. Photoluminescence spectroscopy revealed weak luminescence activity of the materials. Such synthesis of nickel selenides can easily be carried out on a large scale by milling in an industrial mill, as was verified earlier for copper selenide synthesis.

## 1. Introduction

NiSe and NiSe_2_ are members of the VIII-VI binary transition metal chalcogenides group (TMCs). They are p-type semiconductors that scientists are preparing and studying due to their helpful electronic, magnetic, optical, and electrochemical properties applicable in novel smart technologies. NiSe can be used in optical recording materials, optical fibers, sensors, solar cells, laser materials, and conductivity fields [1]. It has also been proposed as a platinum-free catalyst for hydrogen production [2]. In the field of energy storage, NiSe can be a component of Li-ion cells [3]. On the other hand, NiSe_2_ presents promising cathode material for future rechargeable Li-ion batteries [4,5] and recently also the anode material for Na-ion batteries [6]. Methods such as solvothermal [3], hydrothermal [7,8,9], single-source precursor method, thermolysis [1], heating with subsequent milling of the product [10], and polyol technique [10] have so far been used for the synthesis of NiSe. Moreover, different morphologies of NiSe, such as spheres, tubes, nanowires, and flower-shaped structures have been obtained [3,8].

NiSe_2_ thin film was prepared by pulsed laser deposition [4], and cubic particles and nanofibers of NiSe_2_ were synthesized by one-step solvothermal-reduction [5] and electrospinning process [6]. Zhang and co-workers used a new green acetate-paraffin route for NiSe_2_ synthesis [11]. In 2003, Campos and co-authors reported that the problem with the synthesis of binary nickel selenides alloys which have a substantive difference between the melting point of Ni (1455 °C) and Se (221 °C), could be overcome using the mechanical alloying technique [12]. They successfully obtained nanocrystalline NiSe_2_ in the SPEX shaker mill after 10 h of milling.

In terms of a crystal structure, NiSe usually has the hexagonal structure of NiAs. However, the thermodynamically stable rhombohedral structure with a greater anisotropism may also exist. NiSe_2_ crystallizes in the cubic-FeS_2_ type structure, which has dumbbell-shaped Se_2_ units between two Ni atoms [9].

Our several papers and a book chapter deal with the preparation of binary metal selenides and TMCs by the prospective mechanochemical synthesis that can be carried out not only by milling in a laboratory but also in an industrial mill [13,14,15,16]. This research describes the hitherto unreported simple mechanosynthesis of nickel mono-NiSe and diselenide NiSe_2_ by high-energy milling in a planetary ball mill. Both products were characterized from a structural and morphological point of view; their electrical and optical properties were measured and compared.

## 2. Materials and Methods

Mechanosynthesis of NiSe and NiSe_2_ was performed in the planetary ball mill Pulverisette 6 (Fritsch, Idar-Oberstein, Germany) by the milling of nickel (98.8%, 16 μm, Penta Chemicals Ltd., Prague, Czech Republic) and selenium (99.5%, 74 μm, Aldrich, Darmstadt, Germany) powders according to these reactions:(1)$$\mathrm{Ni}+\mathrm{Se}\to \mathrm{NiSe}\;{∆H}_{298}^{0}=-41.9\;\mathrm{to}-77\;\mathrm{kJ}\cdot {\mathrm{mol}}^{-1}$$
(2)$$\mathrm{Ni}+2\;\mathrm{Se}\to {\mathrm{NiSe}}_{2}\;{∆H}_{298}^{0}=-108.9\pm 12.6\;\mathrm{kJ}\cdot {\mathrm{mol}}^{-1}$$

The negative values of enthalpy change of the reactions (1) and (2) reported in Gmelin’s Handbook of Inorganic Chemistry for NiSe_1_._052_ and NiSe_2_ compounds support their thermodynamic feasibility [17]. The milling conditions were as follows. The calculated amounts of Ni powder, 2.13 g and 1.36 g, and Se powder 2.87 g and 3.64 g with a total mass of 5 g for NiSe and NiSe_2_ reactions, were homogenized before milling and put into a tungsten carbide milling chamber with a volume of 250 mL and 50 tungsten carbide balls with a diameter of 10 mm. Subsequently, the milling chamber was filled with an argon atmosphere. The rotation speed was 550 rpm, ball-to-powder ratio 73:1, and milling time from 5 to 120 min. 

X-ray powder diffraction patterns (XRPD) of the milled samples were recorded on a Panalytical Empyrean diffractometer working in θ–2θ Bragg-Brentano geometry using a Cu_Kα1,2_ X-ray source (0.154439 nm). For the incident beam path, 1/4° fixed divergence slit, 1/2° anti-scatter slit, the mask of 10 mm, and Soller slit (0.02 rad) were used. The diffractometer was equipped with a PIXcel3D detector. The phases were identified using the JCPDS PDF database. The data were analyzed utilizing the Fullprof suite software package [18]. For the Rietveld refinement, the diffraction lines were modeled using the Thomas–Cox–Hastings function. During the refinement, lattice parameters, profile parameters, sample displacement, and background points were refined with atomic parameters kept fixed. 

The specific surface area measurements were performed on a Gemini 2360 sorption apparatus (Micromeritics, Norcross, GA, USA) by the low-temperature nitrogen adsorption method. 

The particle size analyzer Mastersizer 2000E with a laser diffraction system (Malvern Pananalytical, Malvern, UK), a dry feeder Scirocco 2000M, and the measuring range 0.02–2000 μm were used for the particle size distributions (PSD) measurements. 

Scanning electron microscopy (SEM) observations were done with the microscope MIRA3 FE-SEM (TESCAN, Brno, Czech Republic), including an EDX detector (Oxford Instrument, Oxford, UK) for the energy dispersive X-ray analysis (EDX) of the observed samples. 

A transmission electron microscopy (TEM) study was performed using a JEOL 2100F UHR microscope operated at 200 kV with a Schottky field emission source. The high-resolution mode was used for taking the images, and the selected area electron diffraction (SAED) was used for the structure identification. Regarding SAED experiments, the microscope was precisely calibrated using MoO_3_ crystal. Gold nanoparticles were used for double-checking, as well. The studied samples were dispersed in absolute ethanol and ultrasonicated for 10 min before observation to reduce the agglomeration of the crystals. These sample dispersions were placed on a copper support grid covered with ultra-thin flat carbon film and stored in a vacuum.

A standard four-point probe technique was used to study the electrical properties of the samples [19,20]. The samples in the form of pellets were pressed from 0.41 g of NiSe and NiSe_2_ powder products under the pressure of 3 t, without retention time, and at room temperature, in a laboratory hydraulic press (Specac, Fort Washington, PA, USA). The four-point test head (Ossila Ltd., Sheffield, UK) was placed on the measured pellet and fixed to its geometrical center during the measurements with a source-measure unit as reported in our previous work [16]. The probe tips were fixed at the same position and loaded with a constant contact force to obtain reproducible results. The distance between the probe tips was 1.27 mm. The diameter of the circular NiSe and NiSe_2_ pellets was 7.06 ± 0.01 mm, and their average room temperature densities of 7.2 and 6.88 g.cm^−3^ were taken from the commonly available literature.

The UV–Vis spectrophotometer Helios Gamma (Thermo Electron Corporation, Warwickshire, UK) was used to measure the optical absorption spectra of the samples. The studied samples were dispersed in absolute ethanol by ultrasonic stirring and poured into the quartz cell. 

A photon-counting spectrofluorometer PC1 (ISS, San Antonio, TX, USA) recorded photoluminescence spectra (PL) of the samples at room temperature and with a photoexcitation wavelength of 360 nm. The excitation source was a 300 W xenon lamp, and the widths of the excitation and emission slits were set to 0.5 and 1 mm. The studied samples were dispersed in absolute ethanol by ultrasonic stirring and placed into the quartz cell cuvette for spectral analysis. 

## 3. Results and Discussion

The results of XRPD indicate that the mechanochemical reaction of Ni with Se according to Equation (1), starts with the formation of the NiSe phase (JCPDS 075-0610), which is already present in the system after 5 min of milling (see Figure 1). After the additional 15 min of milling, the product mainly shows the diffraction peaks belonging to the hexagonal NiSe (sederholmite, *P*6_3_/*mmc* space group) with cell parameters *a* = 3.66 Å nm, *c* = 5.33 Å. Only traces of unreacted elemental nickel metal (JCPDS 004-0850) were detected. Moreover, a weak reflection at 16.5° 2θ might be assigned to the monoclinic wilkmanite Ni_3_Se_4_ phase (JCPDS 018-0890, *I*2/*m* space group), which would be in accordance with the observation of unreacted nickel metal. After 30 min of milling, the weight ratio of the NiSe/Ni_3_Se_4_ phases was determined to be approximately 90/10. The graphical output of the Rietveld refinement in Figure 2 proved that the refined lattice parameters of the NiSe phase are as follows: *a* = *b* = 3.6278(2) Å, *c* = 5.3318(4) Å. The average NiSe crystallite size of 10.5 nm was calculated using the Scherrer formula corrected for instrumental broadening. For the calculation, the integral breadth of (101), (102), (110), and (103) crystallographic planes were used. The provided value is rather informative due to a strong overlap of reflections, high FWHM values, and admixture formation.

The XRPD patterns depicted in Figure 3 confirm the start of the mechanochemical reaction of Ni with 2 Se according to Equation (2) with the formation of the NiSe_2_ (JCPDS 011-0552) already after 10 min of milling as well as for NiSe mechanochemical synthesis, following the more negative formation enthalpy compared to the hexagonal NiSe phase. However, the unreacted nickel (JCPDS 004-0850) and selenium (JCPDS 073-0465) peaks clearly predominate. After 90 min of milling, the penroseite NiSe_2_ phase with the cubic structure (*Pa*-3 space group) and the cell parameter *a* = 5.9604 Å as the major product crystallizes. Only traces of unreacted nickel are present in the pattern. The Rietveld refinement of the XRPD data of the 120 min sample (Figure 4) confirmed NiSe_2_ phase purity with the refined crystal lattice parameter *a* = 5.960 Å, and the average size of crystallites 13.3 nm. 

The phase structure, composition, and morphology of mechanochemically synthesized NiSe and NiSe_2_ can affect the physical and chemical properties of such uniquely prepared materials.

From the dependences in Figure 5, it follows that the specific surface area, SSA_BET_ of both Ni/Se and Ni/2Se samples increases with the time of milling. The maximum value of 2.39 m^2^.g^−1^ for the Ni/Se sample milled for 30 min indicates the completion of the mechanochemical reaction with the formation of the NiSe phase, while the sample Ni/2Se reaches the maximum SSA_BET_ value of 6.73 m^2^.g^−1^ after 60 min of milling. Subsequently, the mean particle size of this sample decreases from 2.08 to 1.69 μm with increasing milling time, but due to the mutual agglomeration of the particles of the NiSe_2_ product, the specific surface area decreases. It is known from the literature that the maximum value of SS_A_ is reached at the end of the mechanochemical reaction for TMCs, but this does not correspond to the case of NiSe_2_ [15,21,22].

Lower magnification SEM images of the NiSe and NiSe_2_ products in Figure 6a and Figure 7a demonstrated the similar agglomerated grains having an irregular shape with inhomogeneous size distribution. The mapping of Ni and Se elements evidenced their homogeneous distribution in both products (Figure 6b and Figure 7b). The quantitative EDX analyses in Figure 6c and Figure 7c confirmed close to the stoichiometric chemical composition of both nickel selenides with atomic ratios of almost 1:1 and 1:2 for NiSe and NiSe_2_. A certain deviation in the determination of the elemental composition of the products can probably be caused by their surface which is not entirely flat. The crystallinity and the crystallite size of the products were also examined in more detail by TEM analysis. 

The crystallite size of the NiSe and NiSe_2_ samples was calculated based on several images taken in bright field STEM mode to avoid the unwanted diffraction effect. More than 50 crystallites were included in the statistics for each sample. Figure 8b showed the result of the statistics for the sample NiSe in the form of a skewed right histogram. The crystallite size in interval 7–12 nm was observed with the highest frequency. Figure 8a illustrates the agglomerates of crystallites in which some displayed atomic layers. The embedded diffraction pattern confirmed NiSe hexagonal phase with space group *P*6_3_/*mmc* (194). The calculated average crystallite size of sample NiSe_2_ was in the interval of 10–15 nm (Figure 9b), which is a bit more than for sample NiSe. The crystallites of the NiSe_2_ sample are visualized in Figure 9a as agglomerated particles. Most of them have distinguishable atomic planes. Analysis of diffraction pattern proved the cubic NiSe_2_ phase with space group *Pa*-3 (205). Analyses of selected area electron diffraction are consistent with XRPD results. When comparing the two TEM images Figure 8a and Figure 9a, it should be taken into account that the images were not acquired at the same magnification.

As data on the electrical properties of nickel selenides are very rare in the literature, the electrical conductivity, resistivity, and sheet resistance values of mechanochemically synthesized NiSe and NiSe_2_ were investigated and measured. Table 1 shows the resulting values of individual electrical quantities coming from 200 measured values for each product. 

The conductivity of NiSe_2_ is slightly higher than that of NiSe, which is related to the larger crystallite size of NiSe_2_ (10–15 nm). This crystallite and/or grain size effect is even more evident at the resistivity value, which is significantly lower for NiSe_2_ than for NiSe (7–12 nm). The electrical resistivity values at room temperature are of the same order as the values reported for nickel selenide prepared by chemical deposition [23], and the solid-state reaction method [24]. The descriptive statistics were performed from the values measured on 20 samples. High standard deviations for the NiSe sample indicate a problem with reproducing the electrical properties of the sample prepared by mechanochemical synthesis. We note that separate statistics for individual samples provide the expected standard deviation, however with significant variation in their mean values.

Nickel selenides are classified as p-type semiconductors, and their bandgap energy value is 620 nm (2.0 eV). According to the literature, the bandgaps of NiSe and NiSe_2_ are narrower than that of nickel oxide (3.6–4.0 eV) and wider than bulk nickel sulphide (0.4 eV). Therefore, they also become potential candidates for solar cells [10]. Figure 10 shows the measured UV–Vis optical absorption spectra of both mechanochemically synthesized nickel mono- and diselenides. Both nickel selenides absorbed UV–Vis radiation in the entire range from 200 to 1000 nm, which fits well when using the solar spectrum. The slightly smaller particles and nanocrystallites of NiSe likely caused the weak excitonic peak at 218 nm (5.65 eV). 

For NiSe_2_, no excitonic peak was observed in the spectrum. The linear part of the Tauc plots in Figure 11 indicates that the NiSe and NiSe_2_ semiconductors involve a direct optical transition. The bandgaps 1.35 eV for NiSe and 1.9 eV for NiSe_2_ were determined by extrapolating the linear regions of the plots to (αhν)^2^ = 0. The bandgap value of NiSe is red-shifted compared to the value of bulk NiSe as well as the value of 1.61 eV for NiSe thin films prepared by chemical deposition and reported by Hankare and co-authors [23]. The determined value of NiSe_2_ bandgap energy more or less corresponds to the bandgap of bulk nickel selenide (2.0 eV). 

The room temperature PL emission spectra of nickel selenides are shown in Figure 12. Light with a wavelength of 360 nm and corresponding to a photon energy of 3.42 eV was used for irradiation—photoexcitation of the samples. This photon energy is absorbed to excite the transition of electrons from the valence band to the conduction band. 

Regarding the peak position, the situation is similar in both samples. The samples have a broad violet emission around 410 nm (3 eV). However, the emission peak of the NiSe_2_ sample has a slightly lower luminescence intensity than NiSe one. Since the emission of PL radiation is mainly due to the recombination of photo-excited electrons and holes, the lower PL intensity could indicate a lower rate of electron/holes recombination in NiSe_2_. Our results are in accordance with previously published papers [1,8,25]. The emission peak situated at 410 nm may be attributed to the defects. These defects occurring in the nanocrystals are generated as a result of high-energy milling. Nevertheless, both samples exhibit only weak photoluminescence.

Table 2 summarizes literature data on NiSe and NiSe_2_ synthesized by various techniques, including mechanochemical synthesis. By comparing the energy and time requirements of individual preparation methods, mechanosynthesis in the planetary mill is a fast, one step, and takes place at ambient temperature. In addition, it does not require solvents, additional reagents, or subsequent operations such as washing and drying the product. The crystallite sizes of NiSe and NiSe_2_ prepared by our procedure reach low values, which could enable their possible applications listed in Table 2 for nickel selenides synthesized by other conventional methods. However, verification and further research are needed.

## 4. Conclusions

Simple, one-step, solvent-free, ecological, relatively economically undemanding, and possible large-scale mechanochemical synthesis of mono- (NiSe) and diselenide (NiSe_2_) nanostructured semiconductors nickel was carried out. XRPD proved that NiSe and NiSe_2_ products with average crystallite sizes of 10.5 and 13.3 nm were synthesized by milling for 30 and 120 min in the planetary ball mill. Rietveld analysis confirmed the hexagonal structure of NiSe (JCPDS 075-0610, space group *P*6_3_/*mmc*) and the cubic structure of NiSe_2_ (JCPDS 011-0552, space group *Pa*-3). Both products contain irregularly shaped nanoparticles that form approximately 1–2 μm agglomerates, as revealed by SEM observations. Qualitative EDX analysis revealed that the atomic ratio of Ni:Se is almost 1:1 and 1:2 for NiSe and NiSe_2_. The results of TEM analyses, SAED patterns indexing, and particle size analyses were in good agreement with the XRPD results. Regarding electrical properties measured for the first time for mechanochemically synthesized nickel selenides, the conductivity of NiSe_2_ was slightly higher, and resistivity was lower than that of NiSe, which is related to the larger crystallite size of NiSe_2_. The calculated energy bandgaps of 1.35 and 1.9 eV for NiSe and NiSe_2_ are in the range suitable for photovoltaics. Their PL spectra showed only a weak emission at 410 nm, which may be attributed to the defects generated by high-energy milling. 

## Figures and Tables

**Figure 1 nanomaterials-12-02952-f001:**
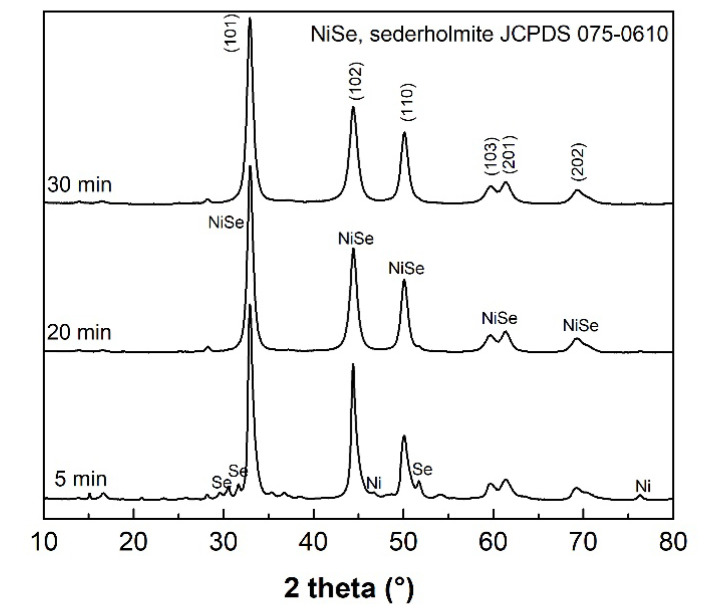
The XRPD patterns of equimolar Ni/Se samples milled for 5, 20, and 30 min.

**Figure 2 nanomaterials-12-02952-f002:**
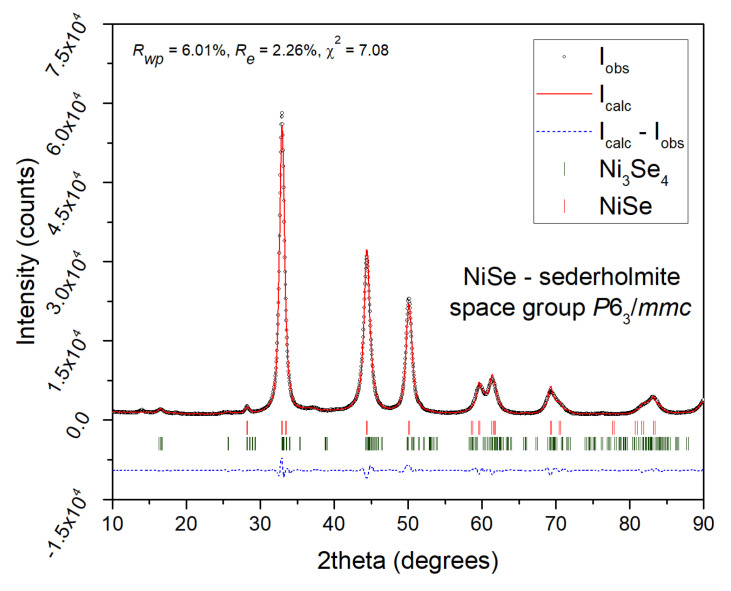
Rietveld refinement graphical output of NiSe—sample milled for 30 min.

**Figure 3 nanomaterials-12-02952-f003:**
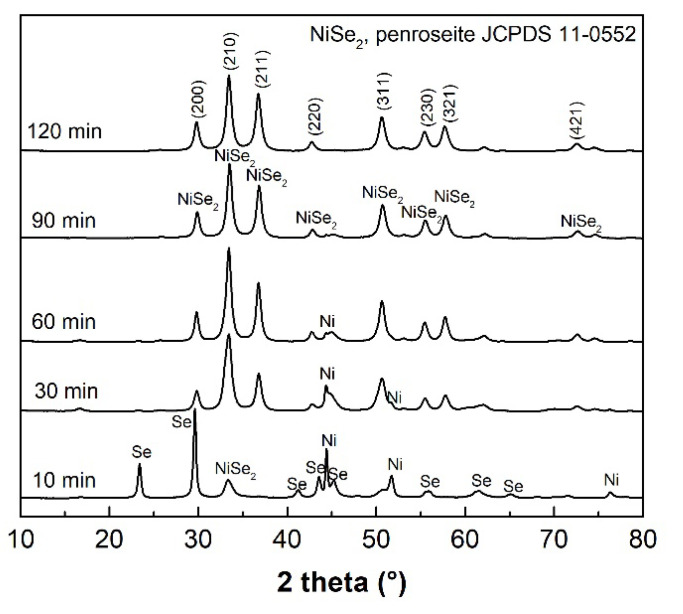
XRPD patterns of Ni/2Se samples milled for 10, 30, 60, 90, and 120 min.

**Figure 4 nanomaterials-12-02952-f004:**
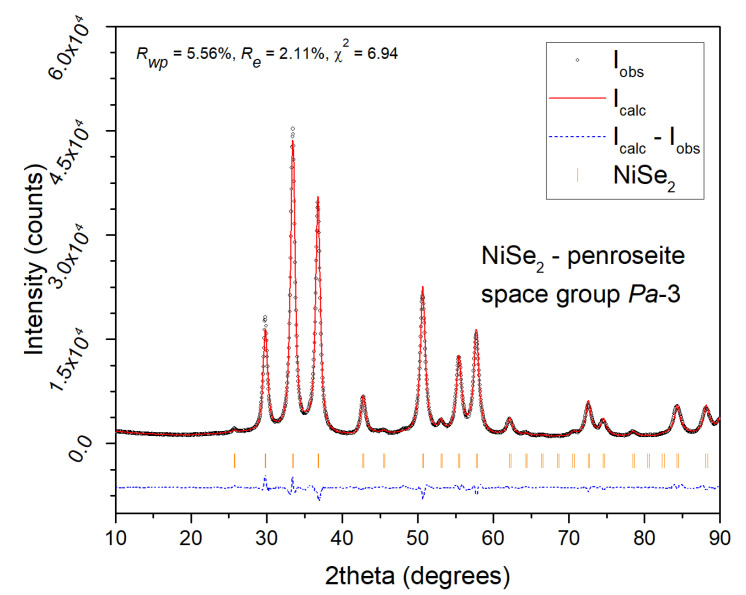
Rietveld refinement graphical outputs of NiSe_2_—sample milled for 120 min.

**Figure 5 nanomaterials-12-02952-f005:**
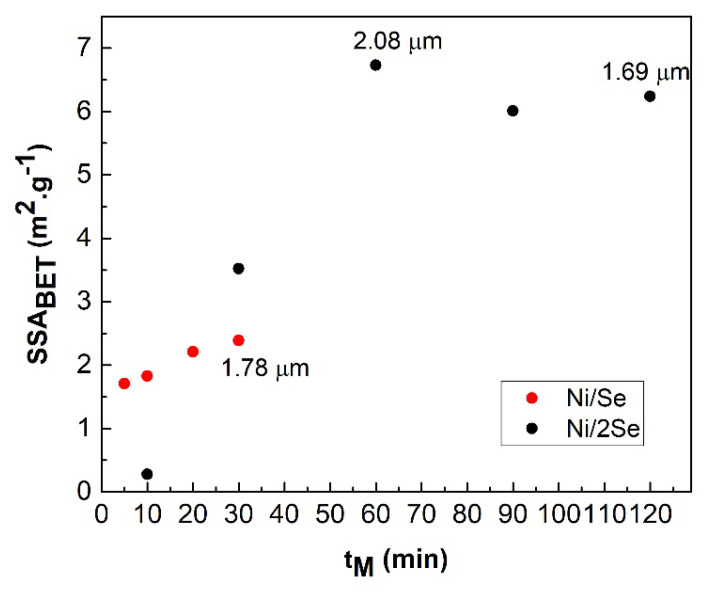
Dependences of specific surface area, SSA_BET_ of Ni/Se and Ni/2Se samples on the time of mechanochemical reaction, t_M_ with the given numerical values of the mean particle size.

**Figure 6 nanomaterials-12-02952-f006:**
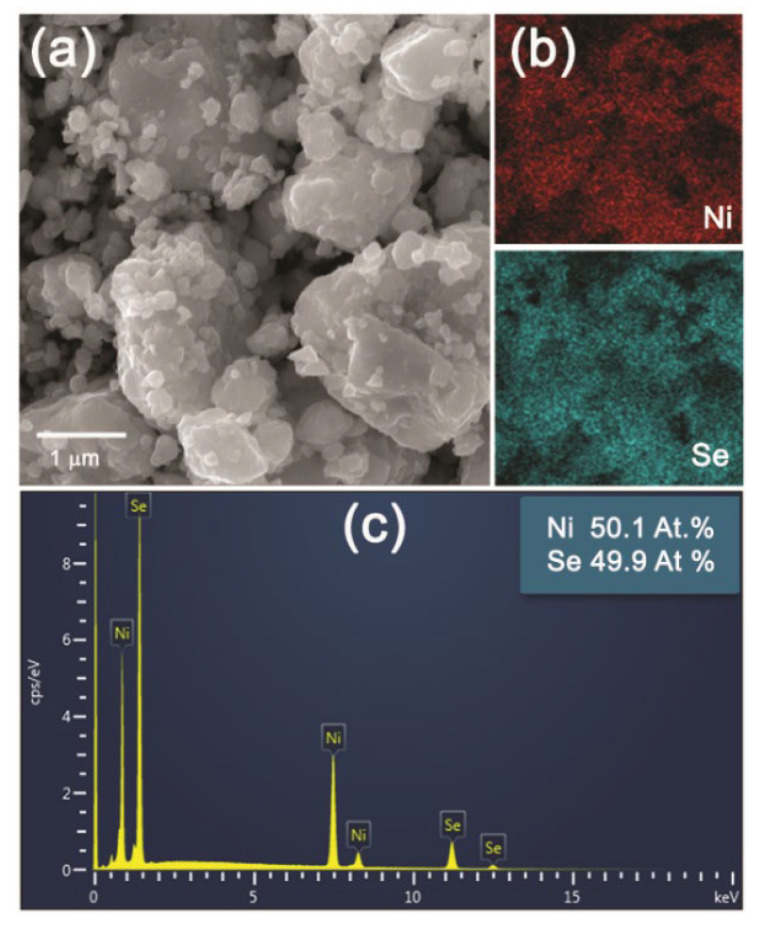
SEM microphotographs of NiSe (**a**), the element mapping for Ni and Se (**b**), and EDX analysis (**c**).

**Figure 7 nanomaterials-12-02952-f007:**
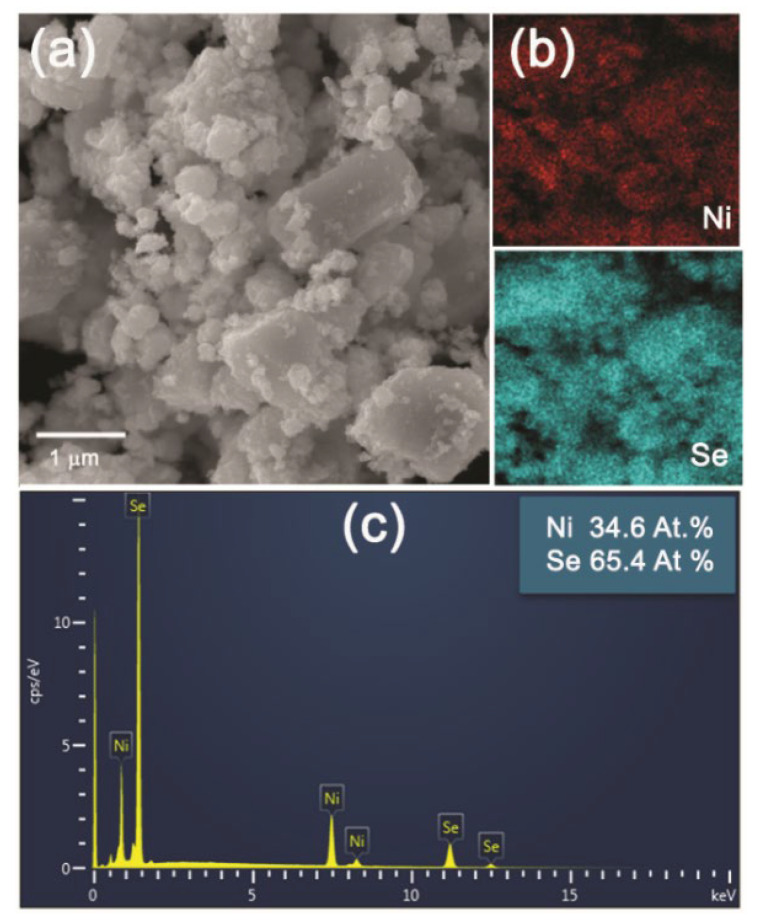
SEM microphotographs of NiSe_2_ (**a**), the element mapping for Ni and Se (**b**), and EDX analysis (**c**).

**Figure 8 nanomaterials-12-02952-f008:**
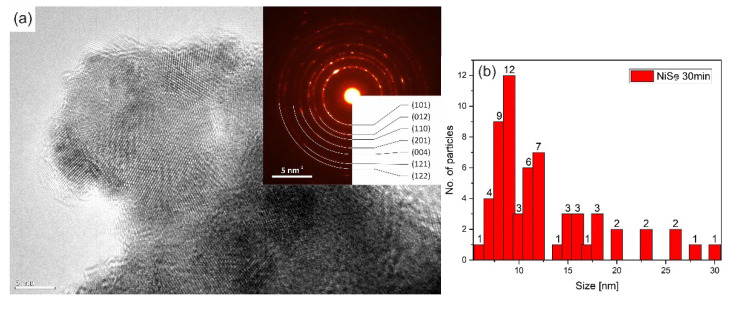
High-resolution TEM micrograph of NiSe agglomerated crystals—magnification 400 kx (**a**) with inset SAED pattern, particle size distribution analysis from TEM observation (**b**).

**Figure 9 nanomaterials-12-02952-f009:**
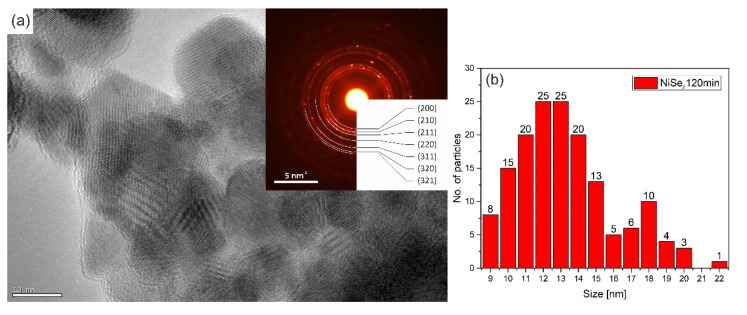
High-resolution TEM micrograph of NiSe_2_ agglomerated crystals—magnification 300 kx (**a**) with inset SAED pattern, particle size distribution analysis from TEM observation (**b**).

**Figure 10 nanomaterials-12-02952-f010:**
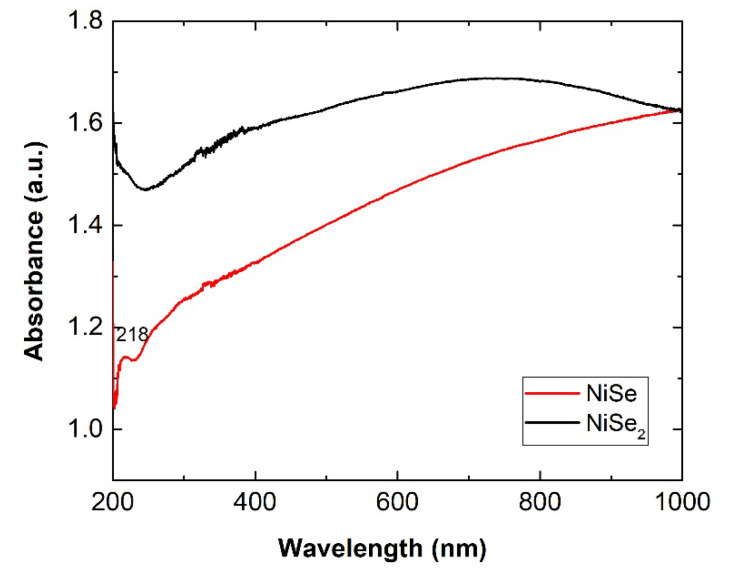
Mono- and diselenide optical absorption spectra.

**Figure 11 nanomaterials-12-02952-f011:**
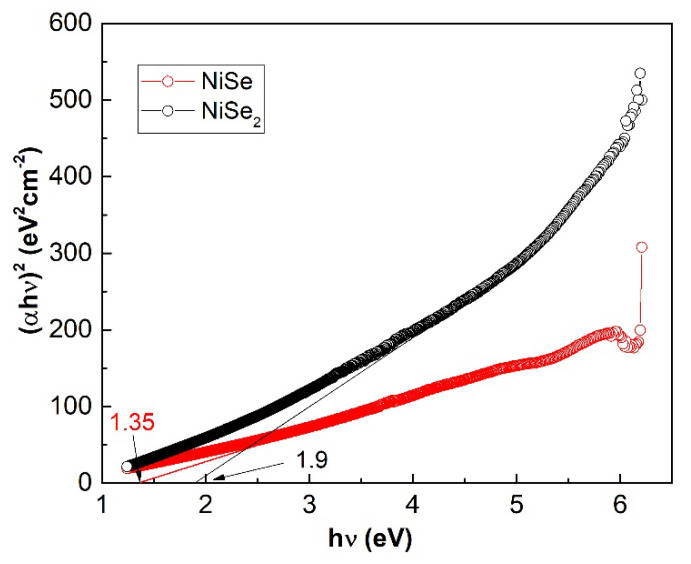
Tauc plots of mono- and diselenide with the obtained bandgaps 1.35 and 1.9 eV.

**Figure 12 nanomaterials-12-02952-f012:**
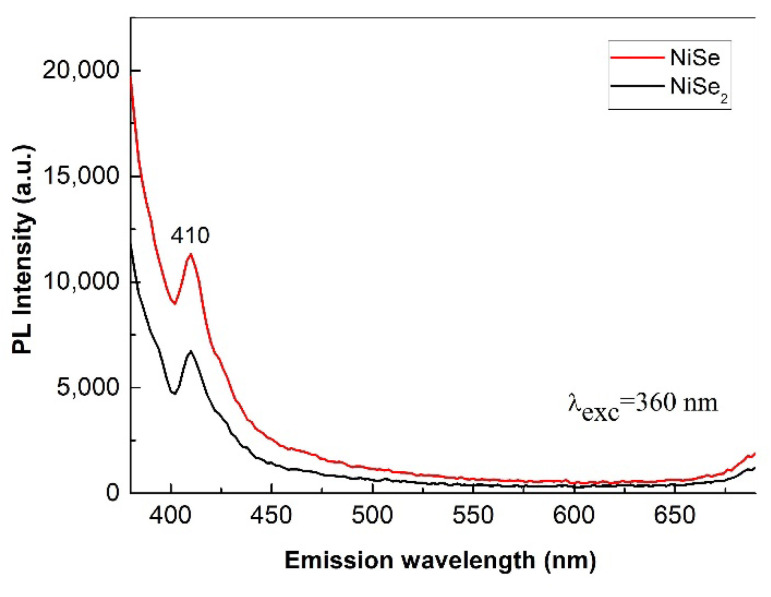
Mono- and diselenide PL emission spectra at an excitation wavelength of 360 nm.

**Table 1 nanomaterials-12-02952-t001:** Electrical quantities of mechanochemically synthesized nickel selenides.

	Product	Mean	Standard Dev	Median
Sheet resistance (mΩ/square)	NiSe	9.34	6.94	7.00
	NiSe_2_	6.26	0.87	6.32
Resistivity (μΩ m)	NiSe	19.98	14.85	14.97
	NiSe_2_	14.43	2.01	14.57
Conductivity (kS/m)	NiSe	68.80	32.82	66.78
	NiSe_2_	70.71	10.42	68.64

**Table 2 nanomaterials-12-02952-t002:** Comparison of the nickel selenides characteristics synthesized by various techniques.

	Method	Temperature/Time	Size	Morphology	Possible Application	Ref.
**NiSe**	Thermolysis	160 °C/1 h	~150 nm	Star-shaped	Solar technology	[1]
	Solvothermal	180 °C/12 h	10–20 nm≤500 nm	Nanofibers	Electrocatalyst for H_2_ production	[2]
	Solvothermal	140 °C/24 h	50 μm	Microtubes	Cathode material for LIBs	[3]
	Polyol	220 °C/3 h	7–9 nm	Nanoparticles	Catalyst- reduction of org. molecules	[10]
	Heating and subsq. milling	960 °C/10 h	7–11 nm	Crystallites	Catalyst: reduction of toxic chemicals	[10]
	Milling in planetary mill	-/30 min	10.5 nm	Crystallites	Unexamined	[this work]
**NiSe_2_**	Pulsed laser deposition		50 nm	Thin film	Cathode material for LIBs	[4]
	Solvothermal reduction	180–280 °C/1 h	30–50 nm	Cubes	Unexamined	[5]
	Electrospinning and subsq. selenization	450 °C/3 h300 °C/10 h	27 nm	Nanofibers	Anode material for SIBs	[6]
	Hydrothermal	140 °C/20 h	3–4 μm	Octahedralcrystals	Unexamined	[9]
	Milling in Spex mill	-/30 h	25 nm	Crystallites	Unexamined	[12]
	Milling in planetary mill	-/2 h	13.3 nm	Crystallites	Unexamined	[this work]

## Data Availability

The data presented in this study are available on request from the corresponding author.

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
