# Peer review of "Mechanochemical Synthesis of Nickel Mono- and Diselenide: Characterization and Electrical and Optical Properties"

_nanomaterials, 2022, doi:10.3390/nano12172952_

Round 1
Reviewer 1 Report
Comments to the authors:
In their contribution, the authors describe a new synthesis strategy for NiSe and NiSe2 via mechanochemistry. The quick room temperature synthesis is well described and the materials are analyzed in-depth. The authors disclose the experimental details excellently and their claims are supported by the results given. The overall work is of good quality but some things have to be changed prior to publishing the results in a suitable fashion.
1) Line 94-96: The sentence: “The specific surface area measurements (BET) were performed on a Gemini 2360 sorption apparatus (Micromeritics, USA) by the low-temperature nitrogen adsorption method.” Has it the wrong way around. You are conducting low-temperature nitrogen adsorption (or physisorption) measurements and then apply the BET equation to the obtained isotherms to obtain the SSABET. Please change this sentence.
2) Line 122: The average room-temperature densities of the samples are given but no indication is given how these values were obtained. Are these literature values, simple calculations including the mass and dimension of the pellet or measured using a machine?
3) Line 140: The authors report the possibility of the formation of Ni3Se4 and provide evidence in the form of leftover Ni and weak reflections. Have they tried to adjust the stoichiometry of the system towards Ni3Se4 to force the system towards this interesting side phase? (Mechanochemical reactions in general have been shown to be very sensitive toward stoichiometry)
4) I don’t understand why the authors decided to combine both Rietveld refinements in figure 2. I would prefer having the refinement for NiSe2 as a separate figure after Figure 3
5) Figure 4 is having several major problems:
a. The connecting lines seem to have no mathematical basis and therefore should be removed. If you have applied a fit in any way no indication is given and the presentation of the data this way suggests, that there is a mathematical relationship between tM and SA which might be true for NiSe but most definitely is not true for NiSe2
b. The legend Ni/Se and Ni/2Se should be corrected to the chemical formulas
c. There is no indication given what the values of 2.08 um are about in the caption
d. The y Axis should be labelled as SSABET (This is true for all occasions where SA is used as abbr. for surface area.
6) Line 180-182: “It is known from the literature that the maximum value of SA is reached at the end of the mechanochemical reaction, but this does not correspond to the case of NiSe2 [15,20,21].” This is not correct. There are ample examples in different material classes where milling leads to smaller particles and in turn higher SSA. Please either specify this to TMCs or leave this statement out.
7) The resolution of Figures 6 and 7 is pretty low. The diffraction pattern is very hard to discern. Please provide a higher resolution.
8) Line 220: The standard deviations given for the NiSe for all investigate quantities are very high, while for the NiSe2 it seems more reasonable. Have there been problems with some of the NiSe samples measured? Can you either exclude some outliers via statistical tests or repeat the measurements to reduce the standard deviation for NiSe?
9) The one thing I am missing in this paper is the analysis of abrasion. The authors utilize WC balls and vessels which in our experience usually produce abrasion. In the Rietveld refinement residuals, one can see minor peaks where WC is expected around 30, 35 and 50° and the EDX also shows a small unassigned peak at around 1.8 eV. Have the authors tried to quantify the abrasion? I think it is vital to address these concerns whenever mechanochemistry is used.
Minor point:
To make it easier for the readers and to confirm with newest trends in mechanochemistry I strongly advise the authors to utilize the reaction scheme and standardized symbols for mechanochemistry as proposed in: https://doi.org/10.3389/fchem.2021.685789
Author Response
- Line 94-96: The sentence: “The specific surface area measurements (BET) were performed on a Gemini 2360 sorption apparatus (Micromeritics, USA) by the low-temperature nitrogen adsorption method.” Has it the wrong way around. You are conducting low-temperature nitrogen adsorption (or physisorption) measurements and then apply the BET equation to the obtained isotherms to obtain the SSAPlease change this sentence.
Reply: Thank you for this comment, we have omitted BET because the equation is a part of the instrument software.
- Line 122: The average room-temperature densities of the samples are given but no indication is given how these values were obtained. Are these literature values, simple calculations including the mass and dimension of the pellet or measured using a machine?
Reply: These are values taken from the available literature. We added it to the sentence in the text of the paper.
- Line 140: The authors report the possibility of the formation of Ni3Se4 and provide evidence in the form of leftover Ni and weak reflections. Have they tried to adjust the stoichiometry of the system towards Ni3Se4 to force the system towards this interesting side phase? (Mechanochemical reactions in general have been shown to be very sensitive toward stoichiometry)
Reply: We have tried to synthesize Ni3Se4 from the precursors in the stoichiometric ratio, but after 60 minutes of milling, in addition to Ni3Se4, NiSe2 is simultaneously formed as well. We will continue to solve this synthesis in the future.
- I don’t understand why the authors decided to combine both Rietveld refinements in figure 2. I would prefer having the refinement for NiSe2 as a separate figure after Figure 3
Reply: We divided Figure 2 into separate Figs. 2 and 4 as you suggested.
5) Figure 4 is having several major problems:
- The connecting lines seem to have no mathematical basis and therefore should be removed. If you have applied a fit in any way no indication is given and the presentation of the data this way suggests, that there is a mathematical relationship between tMand SA which might be true for NiSe but most definitely is not true for NiSe2
Reply: We agree with the reviewer and therefore we removed the connecting lines of the plots.
- The legend Ni/Se and Ni/2Se should be corrected to the chemical formulas
Reply: We kept the legend because these are samples that represent mixtures of precursors that gradually react with milling time.
- There is no indication given what the values of 2.08 um are about in the caption
Reply: Thank you for this notice, we have added to the caption of the Figure: with the given numerical values of the mean particle size.
- The y Axis should be labelled as SSABET(This is true for all occasions where SA is used as abbr. for surface area.
Reply: We have labelled the y-axis as SSABET and corrected it everywhere in the text of the paper.
6) Line 180-182: “It is known from the literature that the maximum value of SA is reached at the end of the mechanochemical reaction, but this does not correspond to the case of NiSe2 [15,20,21].” This is not correct. There are ample examples in different material classes where milling leads to smaller particles and in turn higher SSA. Please either specify this to TMCs or leave this statement out.
Reply: Thank you for this comment, we specified the statement to TMCs.
7) The resolution of Figures 6 and 7 is pretty low. The diffraction pattern is very hard to discern. Please provide a higher resolution.
Reply: Thank you, we have provided higher resolution of these Figures.
8) Line 220: The standard deviations given for the NiSe for all investigate quantities are very high, while for the NiSe2 it seems more reasonable. Have there been problems with some of the NiSe samples measured? Can you either exclude some outliers via statistical tests or repeat the measurements to reduce the standard deviation for NiSe?
Reply: Thank you very much for this comment. The descriptive statistics was performed from the values measured on 20 samples. High standard deviations with NiSe samples indicates a problem with reproducing the electrical properties of the samples using the preparation procedure used. We note that separate statistics for individual samples provide the expected standard deviation, however with significant variation of their mean values. We provide a short commentary on this issue in the revised manuscript.
9) The one thing I am missing in this paper is the analysis of abrasion. The authors utilize WC balls and vessels which in our experience usually produce abrasion. In the Rietveld refinement residuals, one can see minor peaks where WC is expected around 30, 35 and 50° and the EDX also shows a small unassigned peak at around 1.8 eV. Have the authors tried to quantify the abrasion? I think it is vital to address these concerns whenever mechanochemistry is used.
Reply: Due to high FWHM values of individual reflection and strong overlap of both samples with potential WC contamination, we did not include the WC phase in the refinement, we suppose this would lead to negligible improvement of the fit and quantification might not be accurate. Although, the reviewer may be right and especially for high-energy milling with frequently used milling equipment, the presence of WC abrasion can be expected.
Minor point:
To make it easier for the readers and to confirm with newest trends in mechanochemistry I strongly advise the authors to utilize the reaction scheme and standardized symbols for mechanochemistry as proposed in: https://doi.org/10.3389/fchem.2021.685789
Reply: Thank you for this useful reference. We will definitely use it when editing the graphical abstract.
Thank you for the useful comments of the reviewer. We hope that the above will satisfy the reviewer and we sincerely believe that this article will attract the interest of researchers working not only in the field of mechanochemistry.
Reviewer 2 Report
This manuscript describes the syntheses of nanoparticulate NiSe and NiSe₂ by planetary ball milling. The particle formation and properties were studied with a variety of methods, including XRPD, UV-Vis, and photoluminescence spectroscopy. Mechanochemical methods have already been used to prepare a range of nanoscale metal oxide particles, but studies with the heavier chalcogenides are rarer, thus this research is welcome. The route to making these particles is not particularly novel, but the nickel selenides have not been well studied in their nanoparticulate forms, so this represents a useful contribution to the literature. In general, the products are well characterized, and the interpretation of their properties is reasonable. Taken as a whole, this work should attract the interest of material scientists and nanochemists. Publication is recommended, subject to the authors’ consideration of a few points.
1) The electrical quantities in Table 1 show much larger relative standard deviations for NiSe than for NiSe₂ (e.g., the sheet resistance is 9.34±6.94 and 6.26±0.87 mΩ/square for NiSe and NiSe₂, respectively). Given that 200 measurements were made for each compound, one might not have expected such large variations for NiSe. Is this a result of a particle size effect, difficulty in measuring, or something else?
2) In Figure 5, the use of (a), (b), and (c) for both NiSe and NiSe₂ is perhaps not the best labeling. It might be clearer to use (a)–(f) to make sure that everything is distinguished clearly.
Usage:
Page 2, line 61: the “perspective” mechanochemical synthesis—what is meant here? “prospective”?
Page 2, line 69: “was taking place” might be better with something like “was performed”
Page 3, line 147: “the provided value is rather informative” does not agree with the second half of the sentence, where several experimental issues are listed (strong reflection overlap, etc.). Do the authors mean that the provided value should be used with caution?
Page 6, line 176: the sentence beginning “Whereas the sample” is a fragment.
Author Response
1) The electrical quantities in Table 1 show much larger relative standard deviations for NiSe than for NiSe₂ (e.g., the sheet resistance is 9.34±6.94 and 6.26±0.87 mΩ/square for NiSe and NiSe₂, respectively). Given that 200 measurements were made for each compound, one might not have expected such large variations for NiSe. Is this a result of a particle size effect, difficulty in measuring, or something else?
Reply: Thank you. The descriptive statistics was performed from the values measured on 20 samples. High standard deviations with NiSe samples indicates a problem with reproducing the electrical properties of the samples using the preparation procedure used. We note that separate statistics for individual samples provide the expected standard deviation, however with significant variation of their mean values. We provide a short commentary on this issue in the revised manuscript.
2) In Figure 5, the use of (a), (b), and (c) for both NiSe and NiSe₂ is perhaps not the best labeling. It might be clearer to use (a)–(f) to make sure that everything is distinguished clearly.
Reply: We preferred to divide this Figure into separate Figs. 6 and 7 to make it completely clear to the readers
Usage:
Page 2, line 61: the “perspective” mechanochemical synthesis—what is meant here? “prospective”?
Reply: Thank you, we have corrected it.
Page 2, line 69: “was taking place” might be better with something like “was performed”
Reply: Thank you, we have corrected it.
Page 3, line 147: “the provided value is rather informative” does not agree with the second half of the sentence, where several experimental issues are listed (strong reflection overlap, etc.). Do the authors mean that the provided value should be used with caution?
Reply: We apologize for the bad use of language. This was reformulated as follows: „The provided value should be used with caution due to a strong overlap of reflections, high FWHM values and admixture formation“.
Page 6, line 176: the sentence beginning “Whereas the sample” is a fragment.
Reply: Thank you for your comments, we have corrected the text.
Thank you for the useful comments of the reviewer. We hope that the above will satisfy the reviewer and we sincerely believe that this article will attract the interest of researchers working not only in the field of mechanochemistry.
Reviewer 3 Report
The manuscript by Achimovičová et al. reports on the mechanosynthesis of NiSe and NiSe2 semiconductors in the form of nanoparticles. The advantage of this approach appears to be simplicity, the short reaction time and the chance of working at room temperature, as other methods suffer from the low melting point of Se.
The paper contains a full general characterization of the synthetized powders, including XRPD (well done analysis, with Rietveld refinements), electron microscopy coupled to optical (bad gap determination and photoluminescence), and electrical (resistance and electrical conductivity) characterizations. The paper is well written.
I have no scientific concerns about this study. The main limitation is that, since the paper reports no application and results are not commented in view of any application, the interest is not for a general audience, but only for the community of NiSe/NiSe2, which is quite limited.
I would suggest to recap, at the end the discussion, the characteristics of NiSe and NiSe2 nanoparticles compared to those obtained with other conventional techniques, to highlight the peculiarity of the materials produced with mechanosynthesis. Some information is already given while commenting the single results, especially with respect to bulk specimens, but a general comment is still missing, also focusing to possible applications.
Few minor corrections:
- title, I would add a colon in place of a comma before characterization
- line 44-45, “ceramic technique with subsequent milling of the product”. Ceramic technique is vague.. and a reference is missing.
- line 56, “NiSe has the usual anisotropic hexagonal structure”. Why “anisotropic”? “Usual” as it is the literature structure? If so I would rephrase without anisotropic.
- line 69, “was taking place”, I would replace with “took place”
- line 90, the fullprof software should be acknowledge citing the proper reference
- line 146, the Scherrer formula applies to a single peak, typically a low-angle high-intense peak. Why 4 peaks? How has the size been extracted? An average of the four values? Why not to exploit a Williamson-Hall approach?
- Table 1, too many significant digits given for conductivity.
Author Response
I have no scientific concerns about this study. The main limitation is that, since the paper reports no application and results are not commented in view of any application, the interest is not for a general audience, but only for the community of NiSe/NiSe2, which is quite limited. I would suggest to recap, at the end the discussion, the characteristics of NiSe and NiSe2 nanoparticles compared to those obtained with other conventional techniques, to highlight the peculiarity of the materials produced with mechanosynthesis. Some information is already given while commenting the single results, especially with respect to bulk specimens, but a general comment is still missing, also focusing to possible applications.
Reply: At your suggestion, at the end of the discussion, we summarized the data regarding the characteristics of NiSe and NiSe2 prepared by different processes and compared them with our results in the form of Table 2. We inserted the following text into the article and also Table 2: “Table 2 summarizes literature data on NiSe and NiSe2 synthesized by various techniques, including mechanochemical synthesis. By comparing the energy and time requirements of individual…..please see the paper”
Few minor corrections:
- title, I would add a colon in place of a comma before characterization
Reply: Thank you for this idea.
- line 44-45, “ceramic technique with subsequent milling of the product”. Ceramic technique is vague.. and a reference is missing.
Reply: We have changed „ceramic technique“ to heating according to the paper that we also cite, so we have added the missing reference as well.
- line 56, “NiSe has the usual anisotropic hexagonal structure”. Why “anisotropic”? “Usual” as it is the literature structure? If so I would rephrase without anisotropic.
Reply: We have rephrased this sentence.
- line 69, “was taking place”, I would replace with “took place”
Reply: We changed it to “was performed“
- line 90, the fullprof software should be acknowledge citing the proper reference
Reply: We have added proper reference as a citation for the Fullprof suite software.
- line 146, the Scherrer formula applies to a single peak, typically a low-angle high-intense peak. Why 4 peaks? How has the size been extracted? An average of the four values? Why not to exploit a Williamson-Hall approach?
Reply: The reviewer is right. While typically Scherrer formula can be used to calculate crystallite size based upon the integral breadth of a single reflection, it is however not limited and can be given as an average for several values of FWHM (each assigned to different crystallographic planes). This results in a slightly more precise value. The FWHM values used for the calculation were derived from whole pattern fitting. Since the paper is not oriented on an in-depth size, strain, or anisotropy determination, we did not exploit the Williamson-Hall plot, though it could have been an option as well.
- Table 1, too many significant digits given for conductivity.
Reply: Thank you. Table 1 has been modified and the format of conductivity values has been corrected.
Thank you for the useful comments of the reviewer. We hope that the above will satisfy the reviewer and we sincerely believe that this article will attract the interest of researchers working not only in the field of mechanochemistry.
Round 2
Reviewer 1 Report
The authors have addressed my remarks sufficiently. I have no other remarks and support the publication in the current state.